# The association between caesarean section delivery and obesity at age 17 years. Evidence from a longitudinal cohort study in the United Kingdom

**Tessa O. Gorman[1,2], Gillian M. Maher** [1,3]*, **Sukainah Al Khalaf[1,4], Ali S. Khashan[1,3]**

**1** School of Public Health, University College Cork, Cork, Ireland, **2** Department of Public Health South West, St. Finbarr's Hospital, Cork, Ireland, **3** INFANT Research Centre, University College Cork, Cork, Ireland, **4** Mohammed Al-Mana College for Medical Sciences, Dammam, Saudi Arabia

\* Gillian.maher@ucc.ie

**Data Availability Statement:** The Millennium Cohort Study Data is accessible online. It is available here: https://doi.org/10.5255/UKDA-Series-2000031.

## Abstract

### Background

Childhood and adolescent obesity are major, preventable public health concerns. Studies to date are inconclusive regarding an association between caesarean section (CS) delivery and offspring obesity, with fewer studies conducted in late adolescence. This study examined the association between CS delivery, with a specific focus on planned CS, and induction of labour and adolescent body mass index (BMI) and body fat percentage (BF%) at age 17 years.

### Methods

Data on 8,880 mother-child pairs from the United Kingdom Millennium Cohort Study were analysed. The exposures were mode of delivery (normal vaginal delivery (VD) (reference), assisted VD, planned CS and emergency CS) and mode of delivery by induction of labour status. Crude and adjusted binary logistic regression and linear regression models were fitted examining BMI and BF% at age 17 years respectively, adjusting for several potential confounders.

### Results

Adolescents born by CS did not have an elevated BMI or BF% compared to those born by normal VD. The fully adjusted results for overweight and obesity in children born by planned CS, compared to VD, were 1.05 (95% CI: 0.86–1.28) and 0.94 (95% CI: 0.72–1.23), respectively. The results were similar for the associations between CS and BF%, and between induction of labour and BMI.

### Conclusion

Overall, this large longitudinal study did not support an association between CS or induction of labour and overweight, obesity or BF%. It is possible that previously reported associations

**Funding:** The author(s) received no specific funding for this work.

**Competing interests:** The authors have declared that no competing interests exist.

are due to residual or unmeasured confounding and/or underlying indications for CS delivery.

## Introduction

Childhood and adolescent obesity are major, preventable public health concerns. Obesity is commonly measured and defined by body mass index (BMI). Though, several studies suggest body fat percentage (BF%) may be a more accurate measure [1]. From 1975 to 2016, the age standardised prevalence of obesity globally increased from 0.8% to 6.7% [2]. Today, being overweight or obese contributes to more deaths than being underweight [3]. Consequently, eight of the United Nations Sustainable Development Goals incorporate tackling obesity [3]. Childhood and adolescent obesity are a strong predictor of adult obesity, resulting in an increased risk of obesity associated morbidity and mortality as well as economic burden on a societal level [4–6]. Hence, identifying contributing factors to adolescent obesity is crucial to implementing effective public health strategies to combat obesity and the associated adverse health outcomes that transcend into adulthood.

Mode of delivery at birth has been identified as one potentially critical period in the life-course that influences the risk of developing obesity [7, 8]. The rate of caesarean section (CS) delivery is rising globally. From 1990 to 2021, the rate of CS increased from 7% to 21% of all births, and this is predicted to increase to 29% by 2030 [9].

Current evidence is inconclusive and inconsistent regarding the association between CS delivery and obesity in childhood and adolescence [10–14]. The main hypothesised mechanism for association is based on differing alterations to the offspring's gut microbiome during CS compared to vaginal delivery (VD) [15, 16]. The initial gut microbiome of infants is influenced by vertical vaginal and intestinal microbial transmission during delivery which is determined by the mode of delivery [16, 17]. This in turn may affect the transfer of obesogenic microbes from mother to baby during delivery and subsequent nutrient uptake predisposing offspring to obesity [15, 18]. Typically, infant microbial colonisation occurs after the onset of labour. Thus, the induction of labour also contributes to the timing and exposure of the infant to maternal vaginal secretions [16]. For example, if a CS is performed after the onset of labour the infant may still be exposed to maternal vaginal secretions which could impact the formation of their gut microbiome. Another hypothesised mechanism is based on the contribution of adrenaline, cortisol and other chemical factors which are released at different concentrations during labour depending on mode of delivery [7, 19]. These factors may alter neonatal immune-neuroendocrine responses affecting metabolism, and thus, influencing childhood obesity [20].

To date, most research on mode of delivery and obesity focuses on birth to five years of age and young adult life (18–28 years), highlighting the need for further research conducted in adolescents. Furthermore, previous research was often limited by lack of adjustment for pre-pregnancy maternal BMI [13, 21–29]. For example, results of a recent meta-analysis (including 143,416 participants from 13 studies, aged 18–28 years) suggested that CS was associated with a 22% increased likelihood of obesity in young adulthood [26]. However, this result was attenuated when pre-pregnancy maternal BMI was accounted for in a sensitivity analysis, suggesting previous observed associations may be due to unmeasured confounding. Although, more recently, a large prospective birth cohort study conducted in Japan, with 60,769 participants, which adjusted for multiple variables including pre-pregnancy BMI found a positive

association between obesity at 3 years of age when born by CS compared to VD (aRR 1.16 95%, CI 1.08–1.25) [14]. Notably, this study did not disaggregate between emergency and elective CS.

We have previously examined the association between mode of delivery and childhood obesity at ages three, five, seven, eleven and fourteen years using data from the United Kingdom (UK) Millennium Cohort Study (MCS) [10]. The results of this previous study concluded that children (aged three to fourteen years) born by planned or emergency CS did not have a significantly increased BMI compared to those born by normal VD. However, as fewer studies have been conducted in later adolescence it is important to examine the association between CS, and planned CS in particular, and the risk of overweight and obesity in late adolescence because at this age adolescents enter a period of self-transition where physiological, psychological, and social maturation occurs [30]. This transition period is highlighted by the analysis of obesity prevalence in the MCS at 17 years of age, which revealed a sharp increase in the levels of obesity from ages seven to eleven, with the presence of socioeconomic differences in obesity prevalence occurring from age eleven [31]. These additional factors may influence weight change at age 17 years. Therefore, this study aimed to examine the association between CS delivery and adolescent obesity at 17 years of age. The secondary aims were to examine the association between CS delivery and adolescent BF% at 17 years of age and to investigate the contribution of the induction of labour in the association between CS delivery and BMI.

## Methods

### Study design and population

Data were obtained from the United Kingdom (UK) Millennium Cohort Study (MCS). The MCS is a prospective, longitudinal birth cohort study of approximately 19,000 children born in England, Scotland, Wales, and Northern Ireland between 2000 and 2002. The MCS provides comprehensive information on the daily lives, experiences and behaviours of cohort members and their families, alongside anthropometric measures of cohort members. Seven sweeps of data collection have occurred to date, the first sweep of MCS data collection (MCS1) took place when cohort members were nine months old and included 18,818 cohort members (18,552 families). Since then data collections were conducted at ages 3 years (MCS2), 5 years (MCS3), 7 years (MCS4), 11 years (MCS5), 14 years (MCS6) and 17 years (MCS7) [32]. The seventh, most recent sweep of data collection (MCS7) included 10,757 cohort members (10,625 families) [32]. At each study sweep, voluntary informed written consent was obtained from parents for their participation and the participation of their children. Additionally, for MCS7 at age 17 years, verbal consent was obtained from adolescent participants before an interview and physical measurements were performed. Data were collected in the cohort member's home by a computer-assisted interview and procedures were performed by trained interviewers. Full details of the methodologies used can be reviewed elsewhere [33]. Oversampling of certain population subgroups occurred to ensure accurate representation. These subgroups included, cohort members living in Wales, Scotland and Northern Ireland, underserved areas and, areas with higher ethnic minority populations [34]. Cohort members were excluded from the present analysis if there were multiple births (n = 256) or if there was insufficient information provided for birth mode of delivery (n = 81) or BMI at age 17 years (n = 9,681. Ethical approval for the MCS was obtained from the London Multicentre Research Ethics Committee and has therefore been performed in accordance with the ethical standards laid down in the 1964 Declaration of Helsinki and its later amendments. Ethical approval for the secondary analysis was granted by the Social Research Ethics Committee of University

College Cork, Ireland. The data were accessed for research purposes on the 30[th] January 2023. All data were analysed anonymously, with no access to participant identities.

## Exposure

The primary exposure, mode of delivery, was categorised into four groups, normal VD and assisted VD, planned CS and emergency CS. Data were obtained at MCS1 when the child was nine months old through a face-to-face computer assisted personal interview with the child's mother. The child's mother was asked the following question "What type of delivery did you have. Was it a normal delivery, assisted with forceps, assisted vacuum extraction, assisted breach, a planned Caesarean, an emergency Caesarean, or another type of delivery?".

Data were also collected at MCS1 from the cohort member's mother to determine whether the labour was induced. The mother was asked "Was the labour induced or attempted to be induced? (Including any attempt to start labour e.g., injections, pessaries, breaking the waters)". For the induction of labour analysis, the primary exposure was recategorised into eight groups to include induced labour or not induced labour. The MCS did not collate data on the specific indications for CS or induction of labour.

## Outcome

The primary outcome was obesity, defined as body mass index (BMI) in $kg/m^2$ based on the International Obesity Task Force (IOTF) classifications. BMI was categorised as underweight (less than $18.5kg/m^2$), normal healthy weight ($18.5kg/m^2$ to $24.9kg/m^2$), overweight ($25kg/m^2$ to $29.9kg/m^2$) and obese ($\geq 30kg/m^2$). Trained interviewers measured the height and weight of cohort members as per a standardised protocol using a Leicester height measure stadiometer and Tanita scales (BF-522W) respectively. BMI was subsequently calculated as weight divided by height squared. Data were also collected for body fat percentage (BF%). BF% is defined as the total body fat mass divided by total body mass and multiplied by 100. BF% was measured by trained interviewers through foot-to-foot bio-electrical impedance analysis (BIA), using Tanita scales (BF-522W).

## Confounding variables

Data were collected on several potential confounders, as reported in previous literature examining CS-obesity relationship [2, 25–27, 35, 36]. Confounding factors were specified a priori based on the current literature and available variables in the dataset. All confounding factors were measured at MCS1 and grouped into three categories, maternal and cohort member characteristics, maternal health characteristics, and pregnancy complications. Maternal and cohort member characteristics included, maternal age (at time of the cohort members birth), ethnicity, maternal education (highest level achieved), household income (adjusted for household size using the OECD [Organization for Economic Cooperation and Development] schema and classified into quintiles), marital status, fertility assistance, maternal smoking status and alcohol consumption during pregnancy, child gender (at birth), and parity (primiparous/multiparous). Maternal health characteristics included pre-pregnancy BMI (calculated from self-reported height and weight before pregnancy) and diabetes. Pregnancy complications included, pre-term birth (categorised into less than 34 weeks, 34 to 37 weeks and greater than 37 weeks), hypertensive disorder of pregnancy, gestational diabetes and/or raised blood pressure, macrosomia (>4000g), and small for gestational age (defined as birthweight less than the tenth percentile based on maternal reporting of the child's birthweight, gestational age, and gender). Consideration was given to the inclusion of other variables with prior suggested association with childhood obesity, such as breastfeeding, physical activity at 17 years of age and diet at 17 years of age [37, 38].

## Statistical analysis

Descriptive analyses were performed to describe the baseline characteristics of the study population. *Primary analyses*: Crude and adjusted binary logistic regression were performed to estimate odds ratios (OR) and 95% confidence intervals (CI) for the association between CS and obesity in adolescents at age 17 years. To examine the association for each BMI category (underweight, overweight, and obese) compared with normal BMI, binary outcome variables were generated and analysed in separate logistic regression models. Five models were fitted to investigate the influence of multiple confounding factors grouped into three categories, maternal and cohort member characteristics, maternal health characteristics, and pregnancy complications. The five models were; model 1: crude logistic regression analysis, model 2: adjusting for maternal and cohort member characteristics, model 3: adjusting for maternal health characteristics, model 4: (combining models 2 and 3) adjusting for maternal and cohort member characteristics, and maternal health characteristics and, lastly, model 5: fully adjusted model for maternal and cohort member characteristics, maternal health characteristics and pregnancy complications. As it was difficult to determine whether pregnancy complications were potential mediators or potential confounders of the association between mode of delivery and BMI, they were analysed separately in model 5. The reference category for the primary analysis was normal (unassisted) VD and the base outcome was normal BMI.

## Secondary analyses

Crude and fully adjusted linear regression models (models 1 and 5 respectively) were used to examine the association between mode of delivery and BF% at age 17 years. Additionally, to explore the contribution of induction of labour, a new variable with categories for each combination of induction of labour and mode of delivery was created. Again, the crude and maximally adjusted logistic regression models (models 1 and 5 respectively) were fitted for this analysis. The reference category for the mode of delivery and induction analysis was not induced, normal VD and the base outcome was normal BMI. Additional sensitivity analysis was performed excluding cohort members born by planned CS with labour induction, given the possible inaccuracy surrounding this data. Missing data were included as missing data indicators. Post-hoc multinomial logistic regression was also performed examining the association between mode of delivery and BMI category and adjusting for the same variables as in our primary logistic regression model 5. Additionally, post-hoc statistical power analysis was performed. The estimated statistical power provided by the dataset to estimate an OR of 1.3 and 1.5 is based on the observed risk of overweight in the normal VD group of 19%. The estimated risk of overweight in the planned CS group for ORs 1.3 and 1.5 were 25% and 28%, respectively. The statistical power was more than 90% for both. Similarly, the estimated statistical power provided by the dataset to estimate an OR of 1.3 and 1.5 is based on the observed risk of obesity in the normal VD group of 10.5%. The estimated risk of obesity in the planned CS group for the OR of 1.3 and 1.5 were 13.65% and 15.75%, respectively. The statistical power was 73% and more than 90% respectively. All analyses were performed using the statistical software STATA SE/17. Any sensitivity analyses were included in a supporting information file.

## Results

### Descriptive statistics

Out of 18,294 singleton mother-child pairs at baseline, a total of 8,880 continued participation at age 17 years and had data on both mode of delivery and BMI at age 17 years, and therefore

were included in the present analysis. Cohort member characteristics and maternal health characteristics of the participants are outlined in Table 1. Maternal characteristics and pregnancy complications are outlined in S1 Table. The majority of cohort members, 74.3% (n = 6,597) were born by VD; normal VD (68.6%), assisted VD (9.8%). The remaining 25.7% (n = 1,923) were born by CS; planned CS (9.1%), emergency CS (12.6%). 29.8% (n = 2,638) of births were induced, of these 66.3% (n = 3,370) went on to have a normal VD, 12.8% (n = 720) had an assisted VD, 4.1% (n = 231) had a planned CS and 16.8% (n = 949) had an emergency CS. Based on the data the overall IOTF obesity prevalence at 17 years was 10.6% (n = 942).

## Mode of delivery and BMI at age 17 years

In the crude analysis model, adolescents born by emergency CS but not planned CS, had a statistically significant higher odds of obesity at age 17 years compared to those born by normal VD, OR 1.28 (95% CI: 1.05, 1.56) and OR 0.99 (95% CI: 0.77; 1.27) respectively (Table 2). The association between assisted VD and obesity at age 17 did not reach statistical significance, OR 0.78 (95% CI 0.6; 1.01). There was no significant association between mode of delivery and being underweight or overweight in the crude analysis and this persisted across all models.

In model 2, after adjusting for maternal and cohort member characteristics, the association between emergency CS and obesity increased and remained statistically significant, OR 1.42 (95% CI 1.15; 1.76), (Table 2). There was no statistically significant observed association between mode of delivery and any other BMI category. Although, there was a slight increased odds of cohort members born by planned CS being overweight or obese at 17 years, OR 1.15 (95% CI 0.95; 1.4) and OR 1.15 (95% CI 0.89; 1.49), respectively.

In model 3 adjusting for maternal pre-pregnancy BMI and diabetes only, attenuated both the statistically significant association between emergency CS and obesity at 17 years, OR 1.09 (CI 0.88; 1.34), and the increased odds observed between planned CS and overweight and obesity, OR 0.99 (95% CI 0.82; 1.20) and OR 0.80 (95% CI 0.32; 1.04) respectively (Table 2).

In model 4, combining the confounders from models 2 and 3 did not suggest any statistically significant association between mode of delivery and BMI category (Table 2). Notably, adjusting for pre-pregnancy BMI alone accounted for the increased ORs identified in Model 2 (S2 Table).

Similarly, in model 5, the fully adjusted model, following further adjustment for pregnancy complications, mode of delivery was not significantly associated with elevated BMI at age 17 years across any category (Table 3).

Additionally, post-hoc multinomial logistic regression was performed, the results of which support the main conclusions of the study (S3 Table).

## Mode of delivery and BF% at age 17

In crude analysis, there was a statistically significant association between assisted VD and BF% in comparison to the reference group of adolescents delivered by normal VD, crude BF% mean difference -1.21 (95% CI -1.94; -0.48) (Table 3). However, following adjustment for maternal and cohort member characteristics, maternal health characteristics, and pregnancy complications this association no longer persisted, adjusted BF% mean difference -0.28 (95% CI-0.87; 0.32). Other modes of delivery did not show any association with BF% (Table 3).

## Mode of delivery, induction of labour, and BMI at age 17 years

Crude and fully adjusted logistic regression analysis (as per model 1 and 5 respectively) was performed to examine the contribution of induction of labour, to the association between mode of delivery and BMI at age 17 years. In crude analysis, there was a significantly higher

**Table 1. Cohort member characteristics and maternal health characteristics related to mode of delivery among Millennium Cohort Study participants.**

| | Normal VD n (%) | Assisted VD n (%) | Planned CS n (%) | Emergency CS n (%) | Overall n (%) |
|---|---|---|---|---|---|
| | 6,090 (68.6) | 867 (9.8) | 808 (9.1) | 1,115 (12.6) | 8,880 (100.0) |
| **Cohort Member Characteristics** | | | | | |
| Gender at birth | | | | | |
| Male | 2,982 (49.0) | 474 (54.7) | 368 (45.5) | 597 (53.5) | 4,421 (49.8) |
| Female | 3,108 (51.0) | 393 (45.3) | 440 (54.5) | 518 (46.5) | 4,459 (50.2) |
| Gestational age at birth (weeks) | | | | | |
| <37 weeks | 303 (5.0) | 40 (4.6) | 54 (6.7) | 193 (17.3) | 590 (6.6) |
| 37 weeks | 283 (4.7) | 31 (3.6) | 85 (10.5) | 46 (4.1) | 445 (5.0) |
| 38 weeks | 716 (11.8) | 63 (7.3) | 290 (35.9) | 111 (10.0) | 1,180 (13.3) |
| 39 weeks | 1,310 (21.5) | 159 (18.3) | 221 (27.4) | 160 (14.4) | 1,850 (20.8) |
| 40 weeks | 1,907 (31.3) | 294 (33.9) | 90 (11.1) | 265 (23.8) | 2,556 (28.8) |
| >40 weeks | 1,510 (24.8) | 270 (31.1) | 59 (7.3) | 331 (29.7) | 2,170 (24.4) |
| Missing | 61 (1.0) | 10 (1.2) | 9 (1.1) | 9 (0.8) | 89 (1.0) |
| Birth Weight (kg) median IQR | | | | | |
| Median | 3.37 | 3.43 | 3.35 | 3.40 | 3.37 |
| (IQR) | (3.03–3.73) | (3.12–3.77) | (3.00–3.69) | (2.86–3.83) | (3.03–3.74) |
| Labour Induced | | | | | |
| Yes | 1,689 (27.8) | 358 (41.3) | 107 (13.3) | 484 (43.5) | 2,638 (29.8) |
| No | 4,393 (72.2) | 508 (58.7) | 699 (86.7) | 628 (56.5) | 6,228 (70.2) |
| BMI at 17 years | | | | | |
| Underweight | 653 (10.7) | 96 (11.1) | 84 (10.4) | 111 (10.0) | 944 (10.6) |
| Normal | 3,652 (60.0) | 535 (61.7) | 477 (59.0) | 639 (57.3) | 5,303 (59.7) |
| Overweight | 1,143 (18.8) | 163 (18.8) | 164 (20.3) | 221 (19.8) | 1,691 (19.0) |
| Obese | 642 (10.5) | 73 (8.4) | 83 (10.3) | 144 (12.9) | 942 (10.6) |
| Median | 22.2 | 21.9 | 22.2 | 22.5 | 22.3 |
| (IQR) | (20.0–25.4) | (19.8–25.0) | (20.3–25.7) | (20.2–26.0) | (20.0–25.5) |
| Body Fat % at 17 years | | | | | |
| Median | 22.0 | 20.0 | 22.1 | 21.1 | 21.6 |
| (IQR) | (13.9–29.3) | (12.3–27.9) | (14.5–29.7) | (13.1–29.3) | (13.4–29.2) |
| Parity | | | | | |
| Multiparous | 3,933 (64.6) | 186 (21.5) | 588 (72.8) | 396 (35.5) | 5,103 (57.5) |
| Nulliparous | 2,157 (35.4) | 681 (78.5) | 220 (27.2) | 719 (64.5) | 3,777 (42.5) |
| **Maternal Health characteristics** | | | | | |
| Maternal Pre-pregnancy BMI | | | | | |
| Underweight | 317 (5.2) | 38 (4.4) | 22 (2.7) | 46 (4.1) | 423 (4.8) |
| Normal | 3,748 (61.5) | 582 (67.1) | 426 (52.7) | 594 (53.3) | 5,350 (60.3) |
| Overweight | 1,084 (17.8) | 146 (16.8) | 178 (22.0) | 242 (21.7) | 1,650 (18.6) |
| Obese | 420 (6.9) | 52 (6.0) | 98 (12.1) | 136 (12.2) | 706 (8.0) |
| Missing | 521 (8.6) | 49 (5.7) | 84 (10.4) | 97 (8.7) | 751 (8.5) |
| Median | 22.7 | 22.6 | 23.6 | 23.5 | 22.7 |
| (IQR) | (20.6–25.3) | (20.6–24.8) | (21.4–27.1) | (21.4–26.9) | (20.8–25.7) |
| Diabetes Mellitus | | | | | |
| Yes | 6 (0.1) | 3 (0.35) | 5 (0.6) | 1 (0.1) | 15 (0.2) |
| No | 6,083 (99.9) | 864 (99.7) | 801 (99.1) | 1,114 (99.9) | 8,862 (99.8) |
| Missing | 1 (0.0) | 0 (0.0) | 2 (0.3) | 0 (0.0) | 3 (0.0) |

**Table 2. Crude and adjusted logistic regression models examining the association between mode of delivery and BMI category at age 17 years among Millennium Cohort Study participants.**

| Mode of Delivery | Case n | Model 1 [a] OR (95% CI) | p-value | Model 2 [b] OR (95% CI) | p-value | Model 3 [c] OR (95% CI) | p-value | Model 4 [d] OR (95% CI) | p-value | Model 5 [e] OR (95% CI) | p-value |
|---|---|---|---|---|---|---|---|---|---|---|---|
| Normal VD | | ref | | ref | | ref | | ref | | ref | |
| **Underweight** | | | | | | | | | | | |
| Assisted VD | 96 | 1.00 (0.8–1.27) | 0.976 | 1.04 (0.81–1.32) | 0.772 | 1.00 (0.79–1.26) | 0.983 | 1.04 (0.81–1.33) | 0.763 | 1.05 (0.82–1.35) | 0.682 |
| Planned CS | 84 | 0.98 (0.77–1.26) | 0.903 | 1.02 (0.79–1.30) | 0.903 | 1.04 (0.81–1.33) | 0.770 | 1.06 (0.82–1.37) | 0.643 | 1.08 (0.83–1.39) | 0.574 |
| Emergency CS | 111 | 0.97 (0.78–1.21) | 0.795 | 0.96 (0.77–1.21) | 0.758 | 1.02 (0.82–1.27) | 0.890 | 1.01 (0.81–1.27) | 0.902 | 1.03 (0.82–1.31) | 0.780 |
| **Overweight** | | | | | | | | | | | |
| Assisted VD | 163 | 0.97 (0.80–1.17) | 0.779 | 1.02 (0.84–1.25) | 0.814 | 1.00 (0.82–1.21) | 0.977 | 1.02 (0.84–1.24) | 0.849 | 1.02 (0.83–1.24) | 0.883 |
| Planned CS | 164 | 1.10 (0.91–1.33) | 0.331 | 1.15 (0.95–1.40) | 0.154 | 0.99 (0.82–1.20) | 0.914 | 1.05 (0.86–1.28) | 0.633 | 1.05 (0.86–1.28) | 0.650 |
| Emergency CS | 221 | 1.11 (0.94–1.31) | 0.240 | 1.15 (0.97–1.37) | 0.118 | 1.02 (0.86–1.21) | 0.792 | 1.05 (0.88–1.25) | 0.599 | 1.04 (0.87–1.24) | 0.690 |
| **Obese** | | | | | | | | | | | |
| Assisted VD | 73 | 0.78 (0.6–1.01) | **0.055** | 0.89 (0.68–1.17) | 0.392 | 0.82 (0.63–1.06) | 0.133 | 0.88 (0.66–1.16) | 0.356 | 0.87 (0.65–1.15) | 0.321 |
| Planned CS | 83 | 0.99 (0.77–1.27) | 0.935 | 1.15 (0.89–1.49) | 0.270 | 0.80 (0.32–1.04) | 0.095 | 0.95 (0.72–1.24) | 0.688 | 0.94 (0.72–1.23) | 0.653 |
| Emergency CS | 144 | 1.28 (1.05–1.56) | **0.015** | 1.42 (1.15–1.76) | **0.001** | 1.09 (0.88–1.34) | 0.420 | 1.18 (0.95–1.47) | 0.138 | 1.13 (0.90–1.42) | 0.277 |

Abbreviations: VD = vaginal delivery, CS = Caesarean section, CI = confidence interval, OR = odds ratio, ref = reference

[a] Crude analysis

[b] Adjusted for maternal and cohort member characteristics: maternal age/education/smoking/alcohol/ marital status/income/ethnicity, fertility assistance, infant sex, parity.

[c] Adjusted for maternal health characteristics: maternal pre-pregnancy BMI and maternal diabetes

[d] Adjusted for maternal and cohort member characteristics as above and pre-pregnancy maternal BMI and maternal diabetes

[e] Adjusted for maternal and cohort member characteristics and maternal health characteristics as above and pregnancy complications hypertensive disorder of pregnancy, raised blood pressure, gestational diabetes, small for gestational age, preterm and macrosomia.

odds of cohort members who were born via induced normal VD being overweight or obese at age 17 years compared to cohort members born by spontaneous normal VD (not induced), OR 1.26 (95% CI 1.09; 1.45) and OR 1.24 (95% CI 1.03; 1.49) respectively (Table 4). However, there was a statistically significant lower odds of obesity amongst cohort members if not

**Table 3. Crude and adjusted linear regression analysis examining mode of delivery and body fat percent at age 17 years among Millennium Cohort Study participants.**

| Mode of Delivery | Coef. (95% CI) | p-value | Adj Coef (95% CI)[a] | p-value |
|---|---|---|---|---|
| Normal VD | Ref | | Ref | |
| Assisted VD | -1.21 (-1.94; -0.48) | **0.001** | -0.28 (-0.87; 0.32) | 0.368 |
| Planned CS | 0.46 (-0.30; 1.21) | 0.236 | -0.01 (-0.62; 0.59) | 0.965 |
| Emergency CS | -0.24 (-0.90; 0.41) | 0.467 | -0.09 (-0.63; 0.46) | 0.753 |

Abbreviations: VD = vaginal delivery, CS = Caesarean section, CI = confidence interval, Coef. = Coefficient, Adj = adjusted, ref = reference

[a] Adjusted for maternal and cohort member characteristics, maternal health characteristics and pregnancy complications as per Model 5

**Table 4. Crude logistic regression examining the association between mode of delivery including induction of labour and BMI category at age 17 years among Millennium Cohort Study participants.**

| | Underweight | | Overweight | | Obese | |
|---|---|---|---|---|---|---|
| | **OR** | **p-value** | **OR** | **p-value** | **OR** | **p-value** |
| | **(95% CI)** | | **(95% CI)** | | **(95% CI)** | |
| **Mode of Delivery & Induction status** | | | | | | |
| **Normal VD** | | | | | | |
| Not induced | Ref | | Ref | | Ref | |
| Induced | 0.93 (0.77–1.12) | 0.433 | 1.26 (1.09–1.45) | **0.002** | 1.24 (1.03–1.49) | **0.021** |
| **Assisted VD** | | | | | | |
| Not induced | 0.91 (0.67–1.23) | 0.526 | 0.97 (0.76–1.24) | 0.822 | 0.65 (0.45–0.93) | **0.019** |
| Induced | 1.10 (0.78–1.55) | 0.589 | 1.14 (0.86–1.51) | 0.357 | 1.07 (0.74–1.54) | 0.718 |
| **Planned CS** | | | | | | |
| Not induced | 0.95 (0.73–1.25) | 0.731 | 1.18 (0.96–1.45) | 0.115 | 1.04 (0.79–1.36) | 0.785 |
| Induced | 1.04 (0.56–1.95) | 0.896 | 1.14 (0.69–1.87) | 0.617 | 1.06 (0.55–2.02) | 0.872 |
| **Emergency CS** | | | | | | |
| Not induced | 1.07 (0.81–1.41) | 0.646 | 1.15 (0.92–1.43) | 0.23 | 1.47 (1.14–1.90) | **0.003** |
| Induced | 0.81 (0.58–1.14) | 0.22 | 1.21 (0.95–1.54) | 0.117 | 1.21 (0.9–1.64) | 0.211 |

Abbreviations: VD = vaginal delivery, CS = Caesarean section, CI = confidence interval, OR = odds ratio, ref = reference

induced and born by assisted VD, OR 0.65 (95% CI 0.45; 0.93). There was a statistically significant higher odds of obesity amongst cohort members if not induced and born by emergency CS, OR 1.47 (95% CI 1.14; 1.90) (Table 4).

Following maximal adjustment for maternal and cohort characteristics, maternal health characteristics and pregnancy complications, the association between mode of delivery including induction of labour and overweight BMI remained statistically significant for those born by induced normal VD only, OR 1.17 (95% CI 1.0; 1.35) (Table 5). Additional sensitivity analysis was performed on this model excluding cohort members born by induced planned CS and there was no significant change to the findings (S4 Table).

## Discussion

This study examined the association between mode of delivery, with a specific focus on planned CS, and induction of labour and adolescent BMI and BF% at age 17 years. The principal finding indicated that adolescents born by planned or emergency CS did not have an elevated BMI or BF% compared with those born by normal VD. Additional analysis with mode of delivery stratified by induction of labour revealed a statistically significant association between cohort members delivered by normal VD following induction of labour and being overweight at age 17 years, compared to not induced normal VD. However, it is not possible to rule out the role of chance here, especially considering the lack of consistency across the various analyses and in direction of effect.

These results add to the body of evidence refuting the association between CS and elevated offspring BMI and BF%. The results corroborate a similar previous analysis by members of the current study of the association between CS and raised childhood BMI and BF% using MCS data from earlier sweeps at age three, five, seven, eleven and fourteen [10]. This analysis also refutes the causal effect of obesity due to lack of exposure to vaginal microflora at birth in planned CS [28]. Correspondingly, vaginal seeding has been theorised as an intervention to attenuate the differences in the gut microbiome between offspring born by CS and VD [39].

**Table 5. Model 5 adjusted logistic regression examining the association between mode of delivery including induction of labour and BMI category at age 17 years among Millennium Cohort Study participants.**

| | Underweight | | Overweight | | Obese | |
|---|---|---|---|---|---|---|
| | OR | p-value | OR | p-value | OR | p-value |
| | (95% CI) | | (95% CI) | | (95% CI) | |
| **Mode of Delivery & Induction status** | | | | | | |
| **Normal VD** | | | | | | |
| Not induced | Ref | | Ref | | Ref | |
| Induced | 0.96 (0.78–1.16) | 0.649 | 1.17 (1.0–1.35) | **0.045** | 1.06 (0.87–1.28) | 0.56 |
| **Assisted VD** | | | | | | |
| Not induced | 0.92 (0.67–1.26) | 0.605 | 1.04 (0.80–1.34) | 0.775 | 0.78 (0.53–1.14) | 0.195 |
| Induced | 1.23 (0.86–1.77) | 0.256 | 1.10 (0.82–1.48) | 0.517 | 0.98 (0.66–1.46) | 0.939 |
| **Planned CS** | | | | | | |
| Not induced | 1.05 (0.8–1.38) | 0.739 | 1.12 (0.90–1.39) | 0.309 | 0.96 (0.72–1.3) | 0.811 |
| Induced | 1.16 (0.61–2.2) | 0.648 | 0.97 (0.58–1.63) | 0.918 | 0.81 (0.41–1.61) | 0.545 |
| **Emergency CS** | | | | | | |
| Not induced | 1.14 (0.85–1.53) | 0.373 | 1.1 (0.87–1.63) | 0.409 | 1.3 (0.98–1.72) | 0.072 |
| Induced | 0.88 (0.61–1.25) | 0.463 | 1.07 (0.83–1.39) | 0.587 | 0.97 (0.7–1.36) | 0.878 |

Abbreviations: VD = vaginal delivery, CS = Caesarean section, CI = confidence interval, OR = odds ratio, ref = reference

Adjusted for maternal and cohort member characteristics, maternal health characteristics, and pregnancy complications as per Model 5

Vaginal seeding is the exposure of maternal vaginal secretions to a child born by CS, to mimic the transfer of the maternal microbiome during VD. A recent single blind randomised control trial (RCT) evaluating the effects of vaginal seeding on caesarean delivered offspring's BMI found that vaginal seeding had no significant effects on BMI during the first two years of life [40]. The results of this RCT study further undermine the suggestion that obesity occurs in caesarean delivered children due to absence of exposure to vaginal microflora at birth.

The findings from our secondary analysis evaluating the contribution of induction of labour should be interpreted with caution. Particularly as there appears to be a discrepancy in the MCS data, whereby induction of labour occurred, and cohort members were documented as delivered by planned CS. Perhaps these were planned CS arranged before the induction of labour, excluding cohort members that were induced and born by planned CS did not alter our findings (S4 Table).

## Strengths and limitations

There are numerous strengths to this study. Firstly, one of the key strengths is the large, nationally representative cohort of children born in the UK between 2000 and 2002. The prevalence of CS in this study is in keeping with UK national CS rates around the turn of the millennium [9]. Similarly, the prevalence of obesity from our study is in line with the estimated prevalence for obesity applying the IOTF classification in 2019 (8.9% for males and 9.2% for females aged 16 to 24 years) [41]. Secondly, the ability to address significant limitations from previous studies addressing CS delivery and offspring obesity, in particular adjusting for multiple confounders, most notably, maternal pre-pregnancy BMI [35] as well as being able to analyse planned and emergency CS separately [25, 36]. Thirdly, we used internationally standardised and recognised measurements for the outcomes of BMI and BF% at age 17 years, which is a stronger indicator of subsequent BMI in adulthood than BMI in earlier years.

There were also several limitations to the study. First, data on mode of delivery and induction of labour were self-reported by the cohort member's mother at nine months, thus subject to recall bias. However, maternal recall of mode of delivery in the MCS is proven to be highly accurate (94–98%) when compared to hospital records [42]. Conversely, as outlined above, maternal recall for labour induction may be less reliable and subject to misclassification bias [43]. Second, data on potential confounders are subject to recall bias while social desirability bias may also occur, particularly among social matters that are deemed less acceptable e.g., smoking and alcohol consumption during pregnancy [43]. However, maternal reported lifestyle factors such as smoking before and during pregnancy has been shown to be accurate, up to nine years post-delivery when compared with antenatal records on lifestyle factors [44]. It is also worth noting that data on potential confounders were collected nine months post-delivery in the current study and is therefore likely to be more accurate than data collected four to nine years after pregnancy. Third, the effect of unmeasured confounding including confounding due to shared genetics, antimicrobial use during pregnancy and maternal weight gain during pregnancy cannot be ruled out. However, previous studies evaluating CS delivery and childhood weight demonstrated that adjusting for maternal weight gain in pregnancy or antibiotic use in pregnancy did not contribute significantly to the results [29, 45]. Fourth, confounding by indication for mode of delivery may have influenced findings given the reasons behind performing an emergency or planned CS are not known. Fifth, measurement of physical activity and diet within the MCS are limited and therefore were not included in the present analysis. Data on physical activity and diet may be important to consider in future research that suggests an association between CS and overweight/obesity, where possible. Finally, there was potential for selection bias due to missing data and the use of missing data indicators in non-randomised studies can result in biased estimates [46]. However, missing data across mode of delivery was evenly distributed for the main outcome, BMI and therefore is less likely to affect the results (Table 3).

## Conclusion

This study does not indicate an association between CS delivery and an elevated BMI or BF% at age 17 years. Moreover, residual, or unmeasured confounding factors, in particular maternal pre-pregnancy BMI likely account for previously described associations.

## Supporting information

**S1 Table. Maternal characteristics and pregnancy complications characteristics related to mode of delivery among Millennium Cohort Study participants.**
(PDF)

**S2 Table. Logistic regression models examining the association between mode of delivery and obese/overweight BMI at 17 years adjusting for maternal pre-pregnancy BMI and diabetes separately.**
(PDF)

**S3 Table. Multinomial logistic regression examining the association between mode of delivery and BMI category at age 17 years among Millennium Cohort Study participants.**
(PDF)

**S4 Table. Model 5 adjusted logistic regression examining the association between mode of delivery including induction of labour and BMI category at age 17 years excluding induced planned CS.**
(PDF)

## Author Contributions

**Conceptualization:** Tessa O. Gorman, Gillian M. Maher, Ali S. Khashan.

**Formal analysis:** Tessa O. Gorman.

**Methodology:** Tessa O. Gorman, Ali S. Khashan.

**Supervision:** Ali S. Khashan.

**Writing – original draft:** Tessa O. Gorman.

**Writing – review & editing:** Gillian M. Maher, Sukainah Al Khalaf, Ali S. Khashan.

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
