## [Decision Letter · Decision Letter 0]

8 Jan 2024

PONE-D-23-36546The association between caesarean section delivery and obesity at age 17 years. Evidence from a longitudinal cohort study in the United Kingdom.PLOS ONE

Dear Dr. Maher,

Thank you for submitting your manuscript to PLOS ONE. After careful consideration, we feel that it has merit but does not fully meet PLOS ONE’s publication criteria as it currently stands. Therefore, we invite you to submit a revised version of the manuscript that addresses the points raised during the review process.

We look forward to receiving your revised manuscript.

Kind regards,

Gbenga Kayode

Academic Editor

PLOS ONE

Journal Requirements:

Reviewers' comments:

Reviewer's Responses to Questions

**Comments to the Author**

1. Is the manuscript technically sound, and do the data support the conclusions?

Reviewer #1: Yes

Reviewer #2: Yes

2. Has the statistical analysis been performed appropriately and rigorously? 

Reviewer #1: Yes

Reviewer #2: No

3. Have the authors made all data underlying the findings in their manuscript fully available?

Reviewer #1: Yes

Reviewer #2: Yes

4. Is the manuscript presented in an intelligible fashion and written in standard English?

Reviewer #1: Yes

Reviewer #2: Yes

5. Review Comments to the Author

Reviewer #1: This study examined the association between CS and offspring’s obesity at age 17 years using BF% as well as BMI. The analysis revealed the no association, but that does not diminish the importance of this paper. There are several minor comments.

Please calculate the post-hoc power from the sample size and prevalence and include the results. I believe this would further enhance the value of this paper.

Please consider conducting multinomial logistic regression in addition to separate logistic regression analysis as an additional analysis.

Terashita et al. 2023 Sci Rep, 13, 6535, is a recent paper examining the association between CS and child overweight, with using pre-pregnancy BMI as a potential confounder. The authors need to cite this paper.

Reviewer #2: The authors have evaluated the association between caesarean section delivery and obesity at age 17 years. The authors have conducted crude and adjusted logistic regression to estimate the ORs for the association.

Overall comments:

Using data from the same cohort, the authors have previously published on the lack of association between mode of delivery and childhood obesity ate ages 3, 5, 7, 11 and 14 years. Prior studies have shown a strong association between childhood obesity and obesity in later life (adolescence and adulthood). Therefore, I’m not sure how the present manuscript adds any new evidence as the current analyses has been conducted on a subset (i.e. people who had outcome data available at age 17) of the same cohort as their previous study – this is evident in their findings of no association between mode of delivery and obesity at age 17. If there was any association seen at age 17 years that wasn’t evident at age 14 years, then it is highly likely that the association is due to other environmental factors (including lifestyle factors) and not due to mode of delivery.

The authors have fitted five different models to investigate the influence of multiple confounding factors. It is not really clear why some variables were chosen for consideration and others not. Current levels of physical activity and diet have a direct association with current obesity. However, the authors have chosen to omit these variables in their analyses, stating that they were considered to be mediators – are they not effect modifiers? I feel that the selection of variables for model fitting could have been guided by the use of directed acyclic graphs (DAGs).

Other comments:

Table 1 in the manuscript details the characteristics of all the individuals in the cohort. Half of the cohort are missing data on BMI at age 17 years and therefore not included in the analyses. Shouldn’t this table include only the individuals who were included in the analyses?

The authors should note that the use of missing data indicators in non-randomized studies generally results in biased estimates.

The authors state in the Discussion that ‘Additionally, the results support previous research from the MCS and other studies that, maternal pre-pregnancy BMI is a significant contributing factor to offspring obesity’. The analyses reported in the manuscript does not explicitly evaluate the association between maternal BMI and obesity in the child. The fact is that maternal BMI during pregnancy is a confounder as it is associated with mode of delivery and with obesity in the child – the authors have rightly identified it as a confounder a priori and adjusted for in the model.

6. PLOS authors have the option to publish the peer review history of their article (what does this mean?). If published, this will include your full peer review and any attached files.

Reviewer #1: No

Reviewer #2: No

---

## [Author Response · Author response to Decision Letter 0]

21 Feb 2024

22nd February 2024

Dear Gbenga Kayode, 

We thank the editor and reviewers for their helpful feedback on our manuscript entitled “The association between caesarean section delivery and obesity at age 17 years. Evidence from a longitudinal cohort study in the United Kingdom.” We have addressed each comment made by the editor and reviewers. Please see below, in blue, our responses and changes with page and line numbers for the revised manuscript with tracked changes.

Response to Editor

 and 

Author Response: We have made several changes in line with the style requirements outlined.

Author Response: The Millennium Cohort Study Data is accessible online. It is available here: https://doi.org/10.5255/UKDA-Series-2000031

Author Response: The corresponding author is Gillian Maher who is affiliated with University College Cork.

Author Response: Thank you, we have added captions for the Supporting Information files at the end of our manuscript, page 26, lines 510 to 521, which read as follows:

“Supporting information

S1 Table. Maternal characteristics and pregnancy complications characteristics related to mode of delivery among Millennium Cohort Study participants

S2 Table. Logistic Regression models examining the association between Mode of Delivery and obese/overweight BMI at 17 years adjusting for maternal pre-pregnancy BMI and diabetes separately

S3 Table. Model 5 Multinomial Logistic Regression examining the association between Mode of Delivery and BMI category at age 17 years among Millennium Cohort Study participants 

S4 Table. Model 5 Adjusted Logistic Regression examining the association between Mode of Delivery including induction of labour and BMI category at age 17 years excluding induced planned CS”

Response to Reviewers

Reviewer #1: 

This study examined the association between CS and offspring’s obesity at age 17 years using BF% as well as BMI. The analysis revealed the no association, but that does not diminish the importance of this paper. There are several minor comments.

1. Please calculate the post-hoc power from the sample size and prevalence and include the results. I believe this would further enhance the value of this paper.

Author Response: 

Thank you for your suggestion. We calculated the post-hoc power. We included the calculations in the statistical analysis section, page 9 & 10, lines 204 to 212 as follows; 

Statistical Analysis: “Additionally, post-hoc statistical power analysis was performed. The estimated statistical power provided by the dataset to estimate an OR of 1.3 and 1.5 is based on the observed risk of overweight in the normal VD group of 19%. The estimated risk of overweight in the planned CS group for ORs 1.3 and 1.5 were 25% and 28%, respectively. The statistical power was more than 90% for both. Similarly, the estimated statistical power provided by the dataset to estimate an OR of 1.3 and 1.5 is based on the observed risk of obesity in the normal VD group of 10.5%. The estimated risk of obesity in the planned CS group for the OR of 1.3 and 1.5 were 13.65% and 15.75%, respectively. The statistical power was 73% and more than 90% respectively.”

2. Please consider conducting multinomial logistic regression in addition to separate logistic regression analysis as an additional analysis.

Author Response: 

Thank you for your suggestion, we have now conducted multinomial logistic regression as an additional analysis, the results of which support the main conclusions of the study. This is incorporated into our manuscript and Supplementary Table 3 as follows;

Statistical analysis: “Post-hoc multinomial logistic regression was also performed examining the association between mode of delivery and BMI category and adjusting for the same variables as in our primary logistic regression model 5.” Page 9, Lines 202-204, 

Results: “Additionally, post-hoc multinomial logistic regression was performed, the results of which support the main conclusions of the study (S3 Table).” Page 15, lines 259-260

Supporting Information: “S3 Table: Multinomial Logistic Regression examining the association between Mode of Delivery and BMI category at age 17 years among Millennium Cohort Study participants.” Page 26, Lines 516-518

3. Terashita et al. 2023 Sci Rep, 13, 6535, is a recent paper examining the association between CS and child overweight, with using pre-pregnancy BMI as a potential confounder. The authors need to cite this paper.

Author Response: Thank you, we have now cited this paper on page 3 in line 54 and included further reference on page 4, lines 77 to 81. The revised text reads as follows; 

Introduction: “Although, more recently, a large prospective birth cohort study conducted in Japan, with 60,769 participants, which adjusted for multiple variables including pre-pregnancy BMI found a positive association between obesity at 3 years of age when born by CS compared to VD (aRR 1.16 95%, CI 1.08-1.25). [1] Notably, this study did not disaggregate between emergency and elective CS.”

Reviewer #2: 

The authors have evaluated the association between caesarean section delivery and obesity at age 17 years. The authors have conducted crude and adjusted logistic regression to estimate the ORs for the association.

Overall comments:

1. Using data from the same cohort, the authors have previously published on the lack of association between mode of delivery and childhood obesity ate ages 3, 5, 7, 11 and 14 years. Prior studies have shown a strong association between childhood obesity and obesity in later life (adolescence and adulthood). Therefore, I’m not sure how the present manuscript adds any new evidence as the current analyses has been conducted on a subset (i.e. people who had outcome data available at age 17) of the same cohort as their previous study – this is evident in their findings of no association between mode of delivery and obesity at age 17. If there was any association seen at age 17 years that wasn’t evident at age 14 years, then it is highly likely that the association is due to other environmental factors (including lifestyle factors) and not due to mode of delivery.

Author Response:

We appreciate the reviewer’s feedback; however, we think the current manuscript makes a valuable contribution to the field because current evidence is inconclusive and inconsistent regarding the association between CS delivery and obesity in children. Specifically, there is less evidence evaluating the association among adolescents. Assessing obesity at 17 years of age is important because at this age adolescents enter a period of self-transition where physiological, psychological, and social maturation occurs. This transition period is highlighted by the analysis of obesity prevalence in the Millennium Cohort Study at 17 years of age, which revealed a sharp increase in the levels of obesity with age, with the presence of socioeconomic differences in obesity prevalence occurring from age seven to eleven years. These additional factors may be relevant in evaluating the association between caesarean section and obesity at 17 years of age. The present study complements the current evidence available amongst younger children and enhances our understanding of the causes of obesity. Additionally, these findings are important to enable expectant mothers to make evidence based, informed decisions about mode of delivery during pregnancy.

We have edited the manuscript to convey the important contribution this study makes to the field in the following sections:

Introduction: The results of this previous study concluded that children (aged three to fourteen years) born by planned or emergency CS did not have a significantly increased BMI compared to those born by normal VD. However, as fewer studies have been conducted in later adolescence “it is important to examine the association between CS, and planned CS in particular and the risk of overweight and obesity in late adolescence because at this age adolescents enter a period of self-transition where physiological, psychological, and social maturation occurs. [2] This transition period is highlighted by the analysis of obesity prevalence in the MCS at 17 years of age, which revealed a sharp increase in the levels of obesity from ages seven to eleven, with the presence of socioeconomic differences in obesity prevalence occurring from age 11. [3] These additional factors may influence weight change at age 17 years.” Page 5, Lines 87-95

2. The authors have fitted five different models to investigate the influence of multiple confounding factors. It is not really clear why some variables were chosen for consideration and others not. Current levels of physical activity and diet have a direct association with current obesity. However, the authors have chosen to omit these variables in their analyses, stating that they were considered to be mediators – are they not effect modifiers? I feel that the selection of variables for model fitting could have been guided by the use of directed acyclic graphs (DAGs).

Author Response: 

The variables were chosen based on prior literature and careful consideration was given to the included variables by the authors a priori. We have added this to the manuscript on page 7, lines 154-155, which now reads as follows;

“Confounding factors were specified a priori based on the current literature and available variables in the dataset.”

In relation to physical activity and diet, while these variables are captured to a certain extent within the MCS, they are limited, making them difficult to interpret and incorporate into our models. For example, there are questions regarding how many days per week the child does sports, or what they eat and drink between meals. Furthermore, while we believe these are potential mediators, we agree with the reviewer that they could also potentially be effect modifiers. However, considering that we did not find an association between planned CS and overweight or obesity, we feel that it is unlikely to be an issue that they were not included in our analysis. We have now removed the relevant comment in the Confounding variables section (page 8, lines 172 to 173) and accounted for this in the Discussion section (page 22, lines 366 to 369) as follows:

Strengths and limitations: “Fifth, measurement of physical activity and diet within the MCS are limited and therefore were not included in the present analysis. Data on physical activity and diet may be important to consider in future research that suggests an association between CS and overweight/obesity, where possible.” 

Other comments:

3. Table 1 in the manuscript details the characteristics of all the individuals in the cohort. Half of the cohort are missing data on BMI at age 17 years and therefore not included in the analyses. Shouldn’t this table include only the individuals who were included in the analyses?

Authors Response:

Thank you for your suggestion. We have now only included individuals who were included in the analysis. We have edited the Results section (page 10 lines 217 to 228), Table 1 (page 10) and Table S1 (Supplementary Information) to reflect these changes, as follows:

Descriptive Statistics: “Out of 18,294 singleton mother-child pairs at baseline, a total of 8,880 continued participation at age 17 years and had data on both mode of delivery and BMI at age 17 years, and therefore were included in the present analysis. Cohort member characteristics and maternal health characteristics of the participants are outlined in Table 1. Maternal characteristics and pregnancy complications are outlined in S1 Table. The majority of cohort members, 74.3% (n=6,597) were born by VD; normal VD (68.6%), assisted VD (9.8%). The remaining 25.7% (n=1,923) were born by CS; planned CS (9.1%), emergency CS (12.6%). 29.8% (n=2,638) of births were induced, of these 66.3% (n=3,370) went on to have a normal VD, 12.8% (n=720) had an assisted VD, 4.1% (n=231) had a planned CS and 16.8% (n=949) had an emergency CS. Based on the data the overall IOTF obesity prevalence at 17 years was 10.6% (n=942).”

Tables: Table 1 and Supporting Information S1 Table. 

Table 1. Cohort member characteristics and maternal health characteristics related to mode of delivery among Millennium Cohort Study participants

4. The authors should note that the use of missing data indicators in non-randomized studies generally results in biased estimates.

Authors Response: 

We have added the suggested comment to the limitations section, page 22, line 369 to 371, it reads: 

“Finally, there was potential for selection bias due to missing data and the use of missing data indicators in non-randomised studies can result in biased estimates. [4] However, missing data across mode of delivery was evenly distributed for the main outcome, BMI and therefore is less likely to affect the results (Table 3).”

5. The authors state in the Discussion that ‘Additionally, the results support previous research from the MCS and other studies that, maternal pre-pregnancy BMI is a significant contributing factor to offspring obesity’. The analyses reported in the manuscript does not explicitly evaluate the association between maternal BMI and obesity in the child. The fact is that maternal BMI during pregnancy is a confounder as it is associated with mode of delivery and with obesity in the child – the authors have rightly identified it as a confounder a priori and adjusted for in the model.

Authors Response: 

Thank you for your comment, we agree with the reviewer and have omitted this statement from the manuscript (page 20, lines 316 to 317).

We hope these amendments address the suggestions raised. Should any further amendments be necessary, we would be happy to address them accordingly. 

Yours Sincerely,

Dr Tessa O'Gorman, Dr Gillian M. Maher, Dr Sukainah Al Khalaf and Dr Ali S. Khashan

References

1. Terashita S, Yoshida T, Matsumura K, Hatakeyama T, Inadera H. Caesarean section and childhood obesity at age 3 years derived from the Japan Environment and Children's Study. Sci Rep. 2023;13(1):6535.

2. Casey BJ, Duhoux S, Cohen MM. Adolescence: What Do Transmission, Transition, and Translation Have to Do with It? Neuron. 2010;67(5):749-60.

3. Fitzsimons E, Bann D. Obesity prevalence and its inequality from childhood to adolescence: Initial findings from the Millennium Cohort Study Age 17 Survey. London: Centre for Longitudinal Studies.; 2020.

4. Hernán MA, Hernández-Díaz S, Robins JM. A structural approach to selection bias. Epidemiology. 2004;15(5):615-25.

---

## [Editor Report · Decision Letter 1]

20 Mar 2024

The association between caesarean section delivery and obesity at age 17 years. Evidence from a longitudinal cohort study in the United Kingdom.

PONE-D-23-36546R1

Dear Dr. Maher,

We’re pleased to inform you that your manuscript has been judged scientifically suitable for publication and will be formally accepted for publication once it meets all outstanding technical requirements.

Kind regards,

Gbenga Kayode

Academic Editor

PLOS ONE
---

## [Editor Report · Acceptance letter]

22 May 2024

PONE-D-23-36546R1 

PLOS ONE

Dear Dr. Maher, 

I'm pleased to inform you that your manuscript has been deemed suitable for publication in PLOS ONE. Congratulations! Your manuscript is now being handed over to our production team.

Kind regards, 

on behalf of

Dr. Gbenga Kayode 

Academic Editor

PLOS ONE